# Turn-on protein switches for controlling actin binding in cells

Unyime M. Effiong [1], Hannah Khairandish[1], Isabela Ramirez-Velez[1], Yanran Wang[1] & Brian Belardi [1]✉

Within a shared cytoplasm, filamentous actin (F-actin) plays numerous and critical roles across the cell body. Cells rely on actin-binding proteins (ABPs) to organize F-actin and to integrate its polymeric characteristics into diverse cellular processes. Yet, the multitude of ABPs that engage with and shape F-actin make studying a single ABP's influence on cellular activities a significant challenge. Moreover, without a means of manipulating actin-binding sub-cellularly, harnessing the F-actin cytoskeleton for synthetic biology purposes remains elusive. Here, we describe a suite of designed proteins, Controllable Actin-binding Switch Tools (CASTs), whose actin-binding behavior can be controlled with external stimuli. CASTs were developed that respond to different external inputs, providing options for turn-on kinetics and enabling orthogonality and multiplexing. Being genetically encoded, we show that CASTs can be inserted into native protein sequences to control F-actin association locally and engineered into structures to control cell and tissue shape and behavior.

Filamentous proteins of the cytoskeleton drive activities necessary to sustain life, including migration[1–3], adhesion[4–6], endocytosis[7,8], and cell division[9–12]. One example is actin, a highly abundant protein that can form filaments (F-actin) within cells. F-actin adopts multiple higher-order structures[13], and cells rely on these distinct architectures to facilitate many actions. To organize F-actin into different architectures subcellularly, actin-binding proteins bind directly to F-actin, influencing the local kinetics of F-actin assembly and integrating F-actin into multiprotein complexes[14]. In this molecular logic, local abundance and activity of actin-binding proteins regulate how the wide variety of F-actin-dependent processes unfold in living cells.

The largest and most diverse set of actin-binding proteins are those that interface with the sides of actin filaments[15,16]. Within these proteins, actin-binding domains (ABDs) recognize surfaces and grooves built from multiple monomers of actin[17–20]. Proteins possessing ABDs often contain additional domains, linking F-actin side binding to other protein activities. In certain cases, ABDs are combined with sequences that mediate actin-specific functions, such as filament severing and polymerization, motor activity, and multiple filament crosslinking, providing feedback between actin binding and filament

assembly and disassembly[14]. With such a large repertoire of ABD-containing proteins, recent work has focused on the timing and location of single protein engagement with F-actin and the subsequent impact on cell behavior. For instance, the ERM protein, ezrin, was found to associate with membrane-proximal F-actin after phosphorylation of threonine 567, in turn regulating cell cortex mechanics[21,22]. Yet, despite a few notable examples, probing the role of a single actin-binding protein in a multistep cellular process remains a significant challenge due to the lack of tools to control single ABD-F-actin associations.

At present, most methods for manipulating cellular actin cause global perturbation to F-actin structures and binding. Commonly used small molecule drugs and natural products, including latrunculin A and cytochalasin D, disrupt F-actin across the entire cell[23–27], and genetic alterations, such as ABD deletions, lead to constitutive off states[28–31], a major limitation for determining when actin binding is necessary. Several widely used probes exist for visualizing and decorating actin filaments in living cells[32], but imaging and colocalization experiments, unfortunately, do not report on ABD activity. Rather, an approach that controls one ABD against a background of many other

[1]McKetta Department of Chemical Engineering, The University of Texas at Austin, Austin, TX 78712, USA. ✉e-mail: bdb@che.utexas.edu

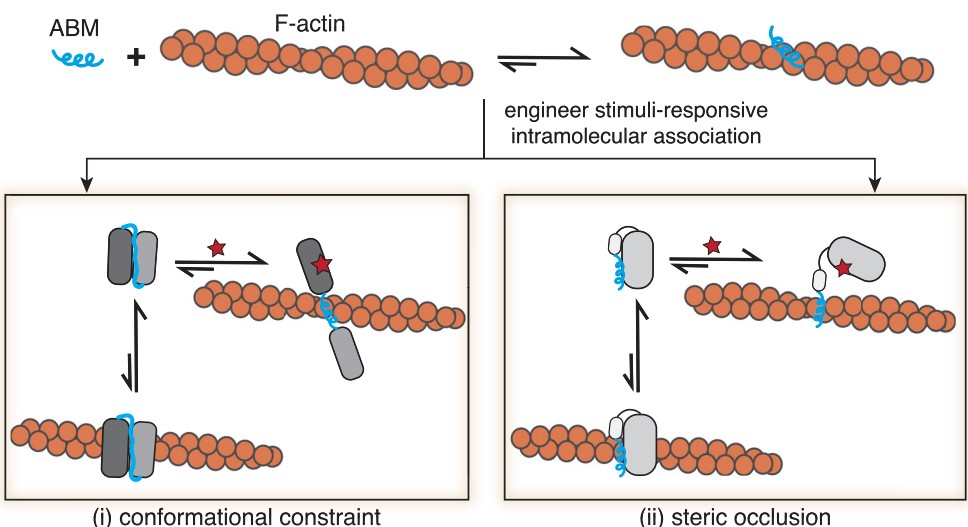

**Fig. 1 | Engineering controllable actin-binding switch tools (CASTs) from actin-binding motifs.** Peptide actin-binding motifs (ABMs) adopt conformations capable of recognizing and binding filamentous actin (F-actin) (top panel), where the bound complexes are energetically favored. To control ABM binding to F-actin, intramolecular binders can be engineered into ABMs' termini (bottom panels), giving rise to Controllable Actin-binding Switch Tools (CASTs). The intramolecular association leads to energetically favorable unbound forms in the presence of F-actin. ABMs can be either (i) conformationally constrained (bottom left) or (ii) sterically occluded (bottom right) to disrupt native F-actin binding. The introduction of a stimulus (red star) that relieves the constraint or steric hindrance would then favor the bound F-actin form of the CAST, turning 'on' binding to F-actin in a user-defined manner.

ABDs would offer the specificity needed for isolating the impact of single ABD-F-actin engagement. As well, manipulating the F-actin association would enable temporal control over F-actin-dependent cellular properties, a goal for synthetic biology applications in eukaryotic cells.

Here, we take advantage of short actin-binding motifs (ABMs) to engineer a suite of Controllable Actin-binding Switch Tools (CASTs) in cells. We and others have recently described native ABMs that consist of short peptide sequences capable of F-actin engagement, including ZO-1's ABS[33], F-tractin[34], and Lifeact[35]. To construct CASTs, our approach relies on installing intramolecular binders on the ends of ABM peptides. By optimizing intramolecular binding, the peptides are forced to adopt constrained or occluded conformations that limit F-actin binding, shifting their association to an inactive state (Fig. 1). The addition of a stimulus, which releases intramolecular binding, then activates the ABM, allowing the sequence to adopt a conformation competent for engaging F-actin. Below, we report on three CASTs that offer different turn-on timescales and orthogonal triggering in living cells. As the described CASTs are genetically encoded, we demonstrate their insertion into multi-domain protein structures to control single protein actin-binding activity in living cells and tissue. We also show that protein switches can be combined with other design elements to generate synthetic actin regulatory proteins that re-organize actin filaments in the presence of a stimulus. These molecular tools provide researchers with additional options[36–39] for studying single protein F-actin association and for engineering cytoskeletal architectures within living cells.

## Results and discussion

### Engineering CASTs to control F-actin binding

Native ABM sequences spontaneously adopt active conformations that engage with F-actin (Fig. 1). Recently solved Cryo-EM structures for the ABM, Lifeact, show an active α-helical structure bound to F-actin at an n/n + 2 site near F-actin's D-loop[18,40]. Active ABM conformations, such as Lifeact's, represent more energetically stable structures in the presence of F-actin compared to inactive states. To construct F-actin protein switches, we sought to re-engineer ABM structures, such that they adopt inactive, yet stable, states. We reasoned that short ABMs

would be more sensitive to structural changes compared to larger ABDs since ABMs lack buffering amino acids or domain architectures, offering an opportunity for rational-based design. To inactivate ABMs, we settled on an approach where intramolecular binders are placed on the N- and C-termini of the ABM sequence. Termini binding would lead to a stable state that conformationally constrains or occludes the ABM, rendering it inactive (Fig. 1). Strong intramolecular binding can then be disrupted by stimuli, such as peptides, small molecules, or light, that relieve the constraint on the ABM and favor the active form of the ABM structure. Here, we focus on engineering two native ABMs, ZO-1's ABS and Lifeact (Supplementary Fig. 1a), the latter a higher affinity ABM sequence toward F-actin and a widely used F-actin marker in cells[35], into CASTs. With our strategy, the ABM's active state would be perturbed in the switch's "off" state, and upon activation, a conformational change would ensue, leading to an unconstrained conformation and resulting in the 'on' state. To directly translate our approach to living cells, we screened CAST candidates in cells using microscopy and co-stained for F-actin to characterize the CAST's properties.

### Peptide-based CASTs for long timescale activation

For our first CAST design, we took advantage of heterospecific synthetic peptides, SynZips (SZs), that form coiled-coil interactions with their peptide partners[41]. SZ4 binds to either SZ3 or SZ21, albeit with a significantly higher affinity for SZ21. We envisioned a design, where SZ3 and SZ4 bind intramolecularly (Fig. 2a), which would disrupt the active form of the ABM constrained between them. Expression of SZ21, the stimulus in this case, would outcompete intramolecular binding, relieving the constraint on the ABM and leading to the F-actin association. Modeling predicted a parallel coiled-coil for the interaction between SZ3 and SZ4 and a C- to N-terminal distance of 7.4 nm (Fig. 2b). This distance must be spanned by the ABM and any adjacent amino acids to allow for intramolecular binding. Since ABMs are short peptide sequences, we considered three different scenarios (Fig. 2c): (i) the ABM length alone does not span the 7.4 nm distance, leaving the ABM in the active state, (ii) a linker sequence on either end of the ABM accommodates the 7.4 nm distance, inactivating the ABM by constraining its conformation, and (iii) additional linker sequences span the required distance for intramolecular binding but do not result in

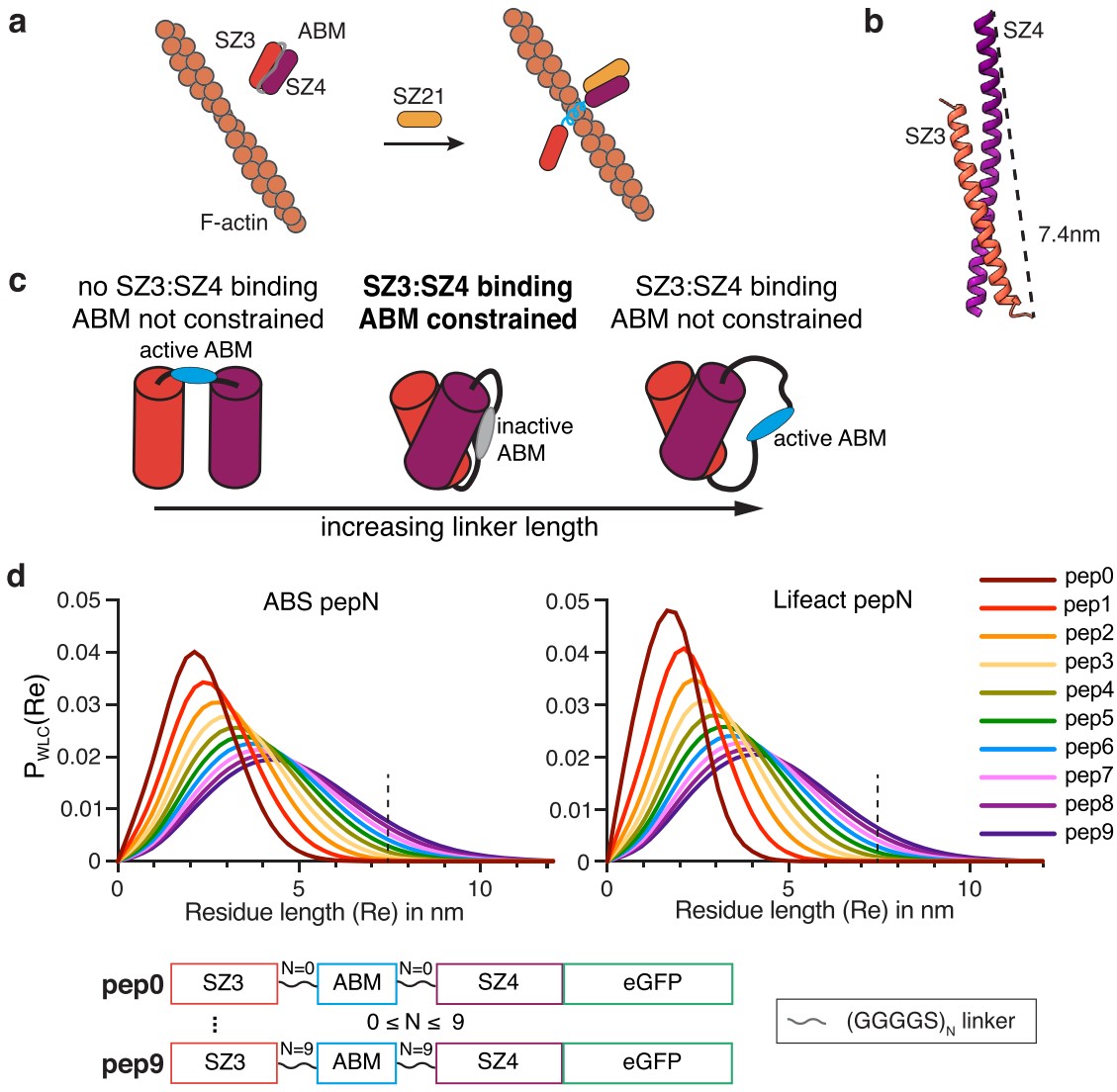

**Fig. 2 | Design of a peptide-responsive CAST. a** Schematic of a peptide-based CAST system. The ABM is initially constrained (gray) in an "off" state by two SynZip sequences, SZ3 and SZ4, which form a coiled-coil interaction intramolecularly. After the introduction of a peptide stimulus (SZ21), which outcompetes SZ3 for SZ4 binding, a transition to the active "on" state (blue) capable of F-actin binding occurs. **b** AlphaFold2 model prediction of SZ3:SZ4 complex with a C-to-N-terminal distance of 7.4 nm. **c** Structural considerations for the initial "off" state of CASTs. The C-to-N-terminal distance of the SZ3:SZ4 complex must be spanned by the interdomain residues. Flexible linkers can be installed on either end of the ABM to increase interdomain length. Short linker lengths impede SZ3:SZ4 binding, allowing the ABM to remain active (left). Long-linker lengths, on the other hand, lead to SZ3:SZ4 binding but no constraint on ABM (right). Only the optimal linker length will allow for both SZ3:SZ4 interaction and a conformational constraint on the ABM (middle), resulting in the "off" state. **d** Probability distributions of end-to-end terminal distances for ABS and Lifeact pep0-pep9 (see bottom) with varying interdomain residue lengths from a worm-like chain (WLC) model (top). The primary sequence of peptide-based CAST candidates for control of F-actin binding (bottom). Source data are provided as a Source Data file.

ABM inactivation. To estimate the amino acid lengths necessary to span the C- to N-terminal distance, we applied a worm-like chain (WLC) model of linker sequences to predict the end-to-end distance distributions[42,43] for both ZO-1's ABS and Lifeact (Fig. 2d). Probabilities over 0.01 were observed for linker lengths with >50 amino acids (AAs). Accordingly, we designed and cloned long-linker SZ-based CAST constructs containing 0–9 five AA-long linkers on both sides of the ABMs, pep0-pep9, for ABS and Lifeact (Fig. 2d).

To examine ABM activity, we expressed pep0-pep9 designs in HeLa cells, which present abundant actin filaments within stress fibers and lamellipodial structures[44,45], and stained for endogenous F-actin using phalloidin (Fig. 3a, c and Supplementary Figs. 1b, 2). We observed similar F-actin engagement for short and long-linker pep designs compared to ABM-only controls, as was anticipated from the structural demands of the SZ3-SZ4 interaction (see above). However, for intermediate linker lengths, we observed robust inactivation of ABM F-actin-binding activity. Using the phalloidin stain as a mask, we compared the fluorescence intensities of the pep0-9 designs on F-actin to the intensities in the cytoplasm and calculated a percent inhibition of F-actin engagement compared to the native ABM's localization (Fig. 3b, d). Designs with >50% inhibition were considered inactive. For both ABS and Lifeact, the pep6 designs, containing 60 AA linker residues, yielded the highest levels of inactivation and were selected for further development (termed pepCAST).

With pepCAST candidates, we turned our attention to the activation of the ABM upon the introduction of SZ21. The presence of SZ21 should disrupt SZ3-SZ4 intramolecular binding and enable ABM engagement with F-actin. To test this, we expressed pepCASTs of ABS and Lifeact in the presence or absence of SZ21. Only in the presence of SZ21 did activation and F-actin engagement of pepCASTs occur

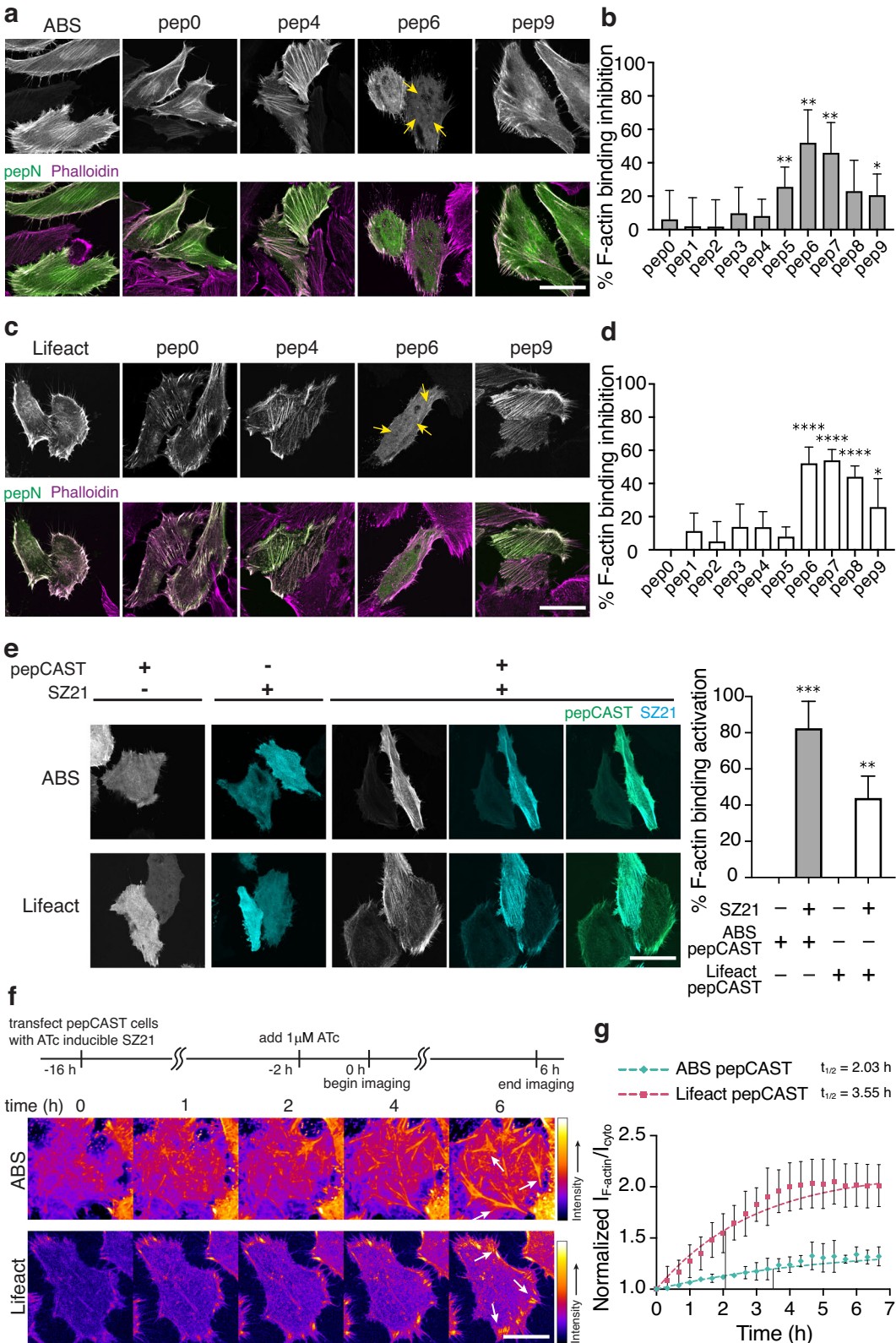

(Fig. 3e), with the ABS pepCAST reaching >80% activation. The stimulus alone, SZ21, was entirely cytoplasmic, but upon dual expression with pepCASTs, SZ21 transitioned to decorating F-actin filaments, consistent with an intermolecular interaction leading to activation of the pepCASTs. We did not observe a correlation between single-cell CAST expression level and activation (Supplementary Fig. 4), suggesting that the concentration of actin far exceeds the concentration

of pepCASTs, giving rise to expression-independent activation. As well, we verified pepCAST activation in vitro with purified forms of pepCAST and F-actin (Supplementary Figs. 5a, 6a, b) and confirmed that pepCASTs do not alter actin polymerization in vitro (Supplementary Fig. 5b).

To determine the kinetics of pepCAST activation, we first generated stable cells expressing either the ABS pepCAST or the Lifeact pepCAST.

**Fig. 3 | Characterization of pepCAST candidates in cells. a** Fluorescent micrographs of fixed HeLa cells expressing ABS pep0-pep9 in the absence of SZ21. Total cellular actin was visualized by Phalloidin staining (magenta). ABS pepCAST (pep6) was selected for further study since its localization to F-actin was significantly reduced (yellow arrows). Scale bar, 30 μm. **b** Quantification of F-actin binding inhibition for ABS pep0-pep9, where 0% is defined by ABS-only localization and 100% is defined by GFP-only localization. Bars represent mean ± SD. $n = 40$ biological replicates. **c** Fluorescent micrographs of fixed HeLa cells expressing Lifeact pep0-pep9 in the absence of SZ21. Total cellular actin was visualized by Phalloidin staining (magenta). Lifeact pepCAST (pep6) was selected for further study since its localization to F-actin was significantly reduced (yellow arrows). Scale bar, 30 μm. **d** Quantification of F-actin binding inhibition for Lifeact pep0-pep9, where 0% is defined by Lifeact-only localization and 100% is defined by GFP-only localization. Bars represent mean ± SD. $n = 40$ biological replicates. **e** Fluorescent micrographs of pepCASTs, SZ21, and the combination of pepCASTs and SZ21 in live HeLa cells

(left). After SZ21 expression, pepCAST localizes to F-actin filaments. Quantification of binding activation (right), where 0% is defined by inactive pepCAST localization and 100% is defined by ABM-only localization. Scale bar, 30 μm. Bars represent mean ± SD. $n = 20$ biological replicates. **f** Time-lapse imaging of pepCASTs shows increased localization of pepCASTs to F-actin over time (white arrows). SZ21 expression was induced with 1 μM ATc to stable pepCAST-expressing HeLa cells transfected with SZ21 in an ATc-inducible plasmid. Scale bar, 30 μm. **g** Kinetics of pepCASTs' activation. The ratio of pepCAST's F-actin-localized fluorescence intensity to its cytoplasmic fluorescence intensity normalized to GFP-only localization is plotted at several timepoints, up to 7 h. $t_{1/2}$ = binding half-time. $n = 3$ biological replicates. Data were fit to a one-phase exponential association model (broken lines). $P$ values were determined using a two-tailed unpaired $t$-test comparison with ABM-only control (**b**, **d**) or inactive pepCAST (**e**). (ns not significant $P > 0.05$; *$P < 0.05$; **$P < 0.01$; ****$P < 0.0001$). Source data are provided as a Source Data file.

With these cells, we expressed SZ21 from an inducible promoter (Fig. 3f). SZ21 expression was observable 2 h after induction with Anhydrotetracycline (ATc) (Supplementary Fig. 3), and we quantified pepCAST F-actin engagement from this time point over the course of 7 h. Both pepCASTs activated as pseudo first-order rate processes on the timescale ($t_{1/2}$) of 2–3 hs (Fig. 3g and Supplementary Movie 1). Our results suggest that intramolecular binding, indeed, leads to inactivity of both ZO-1's ABS and Lifeact, and that F-actin engagement can be restored by disrupting intramolecular binding with a stimulus, in this case SZ21.

## Small molecule-based CAST leads to faster activation

Encouraged by our pepCAST results, we next sought to develop other CAST systems with different turn-on kinetics and stimuli, providing tools that can be used for a variety of actin experiments requiring compatible turn-on times and involving multiple actin-binding proteins. Small molecules as stimuli offer fast turn-on kinetics since they can readily diffuse through the cell membrane and bind to their targets with large rate constants[46,47]. Several small molecules are known to bind tightly to both the catalytically competent and the catalytically dead NS3a protease from the hepatitis C virus and have been used previously to trigger cellular processes[48–51]. To take advantage of these small molecules as stimuli for CASTs, we constructed several possible CAST constructs by fusing catalytically dead NS3a and an NS3a binding ligand (CP5-46A-4D5E, referred to here as CP5)[52], to the termini of the ABMs with and without linkers (Fig. 4a). One arrangement, sm2, caused noticeable disruption to F-actin engagement (Fig. 4b–d). Using AlphaFold2[53,54], we compared a model for the ABS to that of sm2, which we termed smCAST (Fig. 4c and Supplementary Fig. 7). The model indicated that ABS in isolation sits as an α-helix. However, within smCAST, the ABS appeared constrained, where it adopts a disordered structure, a conformational change that would inhibit ABS's F-actin activity when embedded in smCAST.

To examine the activation of smCAST, we incubated smCAST-expressing HeLa cells with either Asunaprevir (Asu), Danoprevir (Dano), or Grazoprevir (Grazo), small molecules known to bind tightly to NS3a's active site[55,56]. In all cases, activation of smCAST and corresponding F-actin engagement occurred, with Asu giving rise to the highest level of activation (Fig. 4e). We verified activation of smCAST in vitro upon the addition of small molecule stimulus (Supplementary Fig. 6c) and confirmed that smCAST does not alter actin polymerization in vitro (Supplementary Fig. 5b). In cells, the extent of smCAST activation was influenced by the concentration of the small molecule stimulus, suggesting a dose-dependent smCAST response (Supplementary Fig. 8). To confirm that activation was due to small molecule binding to NS3a, we cloned two NS3a mutants[57], D182V and D182Y, that do not recognize Asu, in place of NS3a in smCAST. Unlike smCAST, the two variants did not re-localize to F-actin in response to 10 μM Asu (Supplementary Fig. 9), indicating that activation of smCAST is due to direct binding of small molecule stimuli.

To examine smCAST kinetics, we visualized smCAST over time in cells that were treated with different drugs. Similar to pepCASTs, smCAST followed pseudo first-order turn-on kinetics. However, in contrast to pepCASTs, smCAST activated faster than pepCASTs, on the timescale ($t_{1/2}$) of 20 to 30 min, with Asu providing the fastest response of the small molecules tested (Fig. 4f, g and Supplementary Movie 2). This data suggests that varying the intramolecular binders and types of stimuli can lead to additional CAST systems with different turn-on kinetics.

## Rapid activation of light-based CASTs

pepCAST and smCAST activate on hours and tens of minutes timescales, respectively. Still, some cellular processes occur on shorter timescales (single minutes) and, as such, demand more rapid turn-on[58,59]. Light-sensitive protein domains, for example, LOV[60,61] and CRY[62,63] domains, are known to isomerize and conformationally change on shorter timescales[64] than pepCAST's and smCAST's activation. To test whether light can act as a stimulus for CAST activation, we designed optical CASTs, opto1-opto2, by fusing either AsLOV2-Ja' or a mutant with stronger intramolecular association (AsLOV2 L514J-Ja' L531E)[65] to the ABMs. LOV2 domains can sterically occlude short peptide sequences[65,66], which motivated us to examine protein architectures with ABMs at the terminus. For both opto1 and opto2, we observed significant inhibition of F-actin-binding activity, with Lifeact opto1 reaching ~90% inhibition (Fig. 5b, c). The opto1 constructs, termed optoCASTs, were subsequently tested for activation with blue light illumination. This resulted in a rapid and effective redistribution of optoCASTs to F-actin in cells (Fig. 5d), achieving 80% activation, with a $t_{1/2}$ of 3–5 min (Fig. 5e, f and Supplementary Movie 3). Localized activation in individual cells was also possible with optoCAST (Supplementary Fig. 10), showcasing one advantage to a light stimulus. As with pepCASTs and smCAST, purified forms of ABS and Lifeact optoCASTs (Supplementary Fig. 5a) activated in response to blue light illumination in vitro (Supplementary Fig. 6d, e), and the presence of activated optoCASTs did not alter actin polymerization (Supplementary Fig. 5b).

With smCAST and optoCAST—two switches that can be activated within the same order of magnitude—in hand, we next examined whether orthogonal triggering was possible. To do this, we co-expressed smCAST and Lifeact optoCAST in cells and triggered activation with either small molecules, such as Asu, Dano, and Grazo, or blue light. In the presence of small molecule stimuli, activation of smCAST ensued with the re-localization of smCAST to F-actin, while optoCAST remained cytoplasmic. Similarly, blue light stimulus caused optoCAST to decorate F-actin, whereas smCAST remained cytoplasmic. Sequential activation within the same cell was also possible (Supplementary Fig. 11). Together, these experiments show that each CAST is sensitive to its distinct stimulus and that multiple CASTs can be used simultaneously for multiplexed and orthogonal control over F-actin binding in cells.

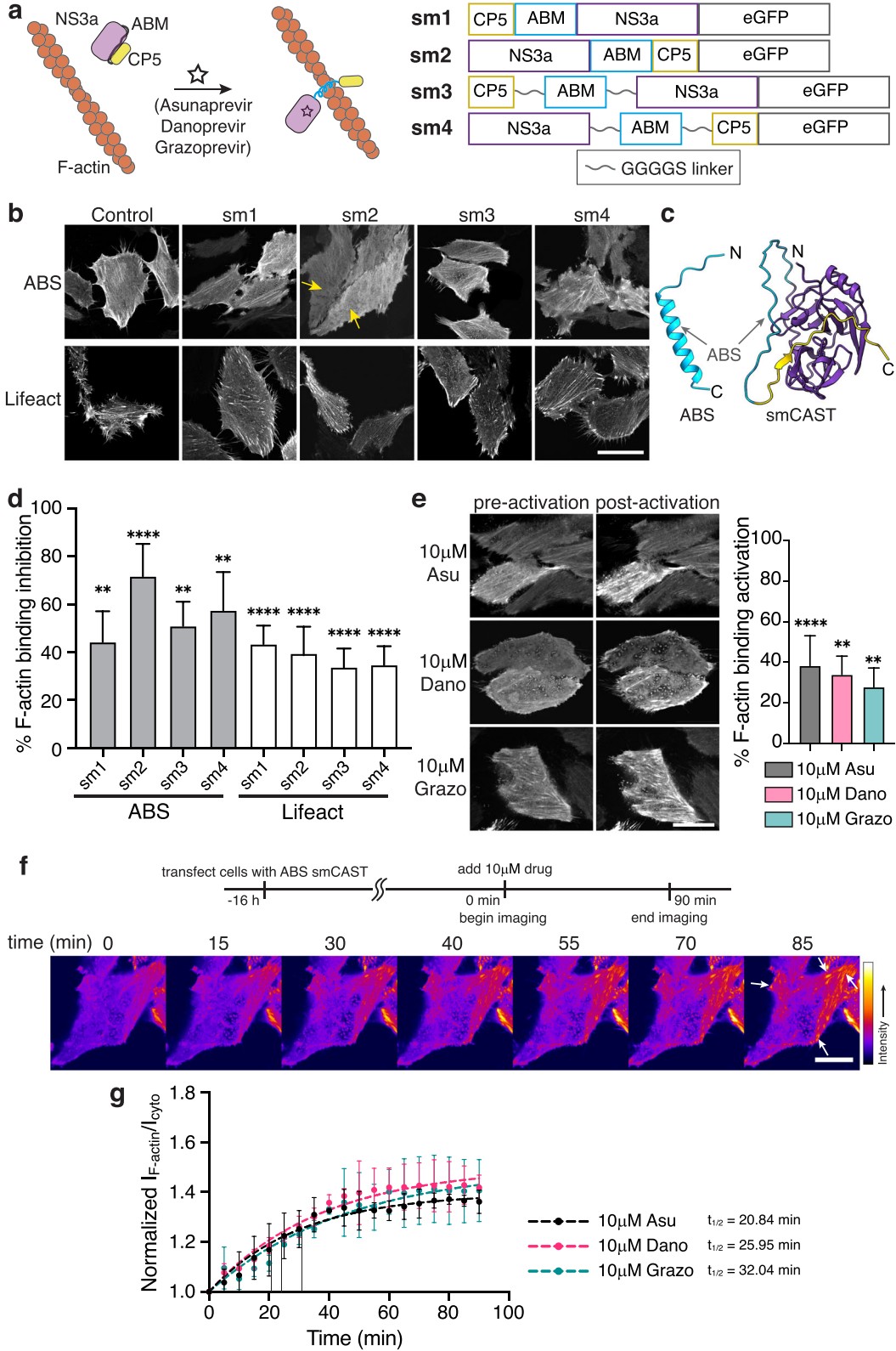

## Dimeric optoCASTs enable control over F-actin architecture and cell shape with light

By varying F-actin's organization, cells regulate their shape, movement, and mechanics. Bundled filaments, for instance, represent a prominent F-actin architecture in stress fibers and at the cell cortex[13,67,68]. CASTs offer an opportunity to induce F-actin re-organization and subsequent cellular responses by engineering the intramolecular-bound states into structures that can alter F-actin architecture dynamically. Dimeric ABD-containing proteins, so-called actin crosslinkers, are responsible for forming bundled filaments in the cell[69–71], and we reasoned that generating a dimeric CAST should provide control over filament bundling in a user-defined manner. To construct multimeric CASTs in cells, we appended a self-dimerizing peptide sequence to the N-terminus of optoCASTs, termed dOptoABS

**Fig. 4 | Characterization of small molecule-based CAST candidates in cells.**
**a** Schematic of a small molecule-based CAST system. The ABM is initially con-strained (gray) by the intramolecular association of NS3a and CP5. After the addition of small molecule inhibitors (Asunaprevir, Danoprevir, or Grazoprevir) that disrupt the NS3a:CP5 complex, the CAST transitions to an active state (blue) capable of F-actin binding (left). The primary sequence of small molecule-based CAST candidates tested (right). **b** Fluorescent micrographs of fixed HeLa cells expressing sm1-sm4 in the absence of small molecule inhibitors. smCAST (sm2) was selected for further study since its localization to F-actin was significantly reduced (yellow arrows). Scale bar 30 µm. **c** AlphaFold2 prediction of native ABS (blue, left) and smCAST (right). ABS alone is predicted to be an α-helix, the active form of the ABM, whereas within smCAST the ABS sequence is unstructured (right). **d** Quantification of F-actin binding inhibition for ABS and Lifeact sm1-sm4, where 0% is defined by ABS-only localization and 100% is defined by GFP-only localization. Bars represent mean ± SD. $n = 40$ biological replicates. **e** Fluorescent micrographs of smCAST in the presence or absence of Asu, Dano, or Grazo in live HeLa cells (left). Quantification of activation in the presence of drug for 30 min (right), where 0% is defined by inactive smCAST localization and 100% is defined by ABS-only localization. Scale bar 20 µm. Bars represent mean ± SD. $n = 30$ biological replicates. **f** Time-lapse imaging of the smCAST after the addition of Grazo shows increased localization of smCAST to F-actin (white arrows) over time. Scale bar 10 µm. **g** Kinetics of smCAST activation. The ratio of smCAST's F-actin-localized fluorescence intensity to its cytoplasmic fluorescence intensity normalized to GFP-only localization is plotted at several timepoints, up to 90 min. $t_{1/2}$ = binding half-time. $n = 3$ biological replicates. Data were fit to a one-phase exponential association model (broken lines). $P$ values were determined using a two-tailed unpaired $t$-test comparison with ABS-only control (**d**) or inactive smCAST (**e**). (ns not significant $P > 0.05$; *$P < 0.05$; **$P < 0.01$; ****$P < 0.0001$). Source data are provided as a Source Data file.

and dOptoLifeact (Fig. 6a), and expressed these sequences in HEK 293 cells, which lack extensive bundled filaments (Supplementary Fig. 12a). After illuminating with blue light, we observed dramatic re-organization of F-actin into heavily crosslinked architectures at both the cell periphery and in the cell body for both dOptoABS and dOptoLifeact (Fig. 6b and Supplementary Movie 4). Coincident with crosslinking, cells also appeared to contract after blue light triggering. Quantification of cell area indicated an area reduction of approximately 10% for dOptoABS and 20% for dOptoLifeact upon activation. Next, we wondered whether triggering actin crosslinking and cell area changes would be possible in more crowded environments, such as in tissues. We, consequently, expressed either dOptoABS or dOptoLifeact in MDCK II epithelial monolayers and activated with blue light. After blue light illumination, we observed similar F-actin re-organization as in isolated cell experiments (Fig. 6c and Supplementary Movie 5), but, in the case of tissue, we also found that cell contacts were disrupted with the majority of cells becoming detached from their neighbors after dimeric CAST activation. These experiments show that CASTs are amenable to further engineering for synthetic biology applications, where control over F-actin organization is desired.

### Manipulating proteins and cellular processes dynamically by replacing native ABDs with CASTs

Finally, we sought to test whether CASTs can be used in place of a native ABD to control protein activity in cells. To do this, we focused on the junctional protein, ZO-1, which plays an active role in focal adhesion formation and cell migration[72,73]. ZO-1, a major component of the tight junction in epithelial tissue, also helps drive persistent migration of subconfluent cells by interacting with integrin α5β1 and regulating focal adhesions[74,75]. We expressed either wildtype ZO-1 (WT ZO-1) or ZO-1 lacking its ABS (ZO-1ΔABS) in subconfluent cells (Fig. 7a) and found that, indeed, WT ZO-1 localizes to focal complexes at the cell periphery and to larger focal adhesion clusters under the cell body (Fig. 7b). In contrast, ZO-1ΔABS primarily localized to small focal complexes on the cell periphery, indicating that ZO-1's engagement with F-actin modulates its adhesion activity. Next, we constructed a version of ZO-1 that replaces ZO-1's native ABS sequence with our ABS smCAST, termed ZO-1smCAST (Fig. 7a). From our data above, we anticipated that if actin-binding of ZO-1 occurred, then the protein would populate larger focal adhesion clusters over time. By fluorescence microscopy, we first imaged ZO-1smCAST and found that it localizes to small focal complexes at the cell periphery (Fig. 7c), similar to ZO-1ΔABS. After activation with either Asu, Dano, or Grazo, however, ZO-1smCAST re-localized over the course of 60 min, occupying larger focal adhesions under the cell body and mirroring that of WT ZO-1 (Fig. 7c and Supplementary Movie 6), (Fig. 7d, e). Encouraged by these results, we then turned to a wound healing assay of collective cell migration. We've previously shown that ZO proteins, including ZO-1, are essential for epithelial cell migration using a knockout line for ZO-1

and ZO-2[73]. To this knockout line, we expressed either WT ZO-1 or ZO-1smCAST and performed a wound healing experiment. WT ZO-1-expressing cells were able to migrate and close the wound area over the course of 24 h, whereas the ZO-1smCAST-expressing cells did not close the wound area after 52 h (Fig. 7f–h), suggesting that ZO-1's ability to bind to F-actin is necessary for collective migration. To test this directly, we repeated the wound healing assay with ZO-1smCAST-expressing cells and added Asu, Dano, or Grazo to activate the smCAST module within ZO-1's structure. Gratifyingly, in all cases, ZO-1smCAST-expressing cells were able to migrate and close the wound area in the presence of small molecule stimuli on timescales similar to WT ZO-1-expressing cells (Fig. 7f–i and Supplementary Fig. 13a–c). Small molecule stimuli, Asu, Dano, or Grazo, had no effect on WT ZO-1-expressing cells. Taken together, our data show that CASTs can be incorporated into the native sequence of proteins to control their engagement with F-actin in both cells and tissue.

### Concluding remarks

F-actin plays numerous roles throughout the cell, and a diverse set of proteins bearing ABDs orchestrate the varied activities of F-actin. Identifying which ABD-containing proteins are necessary for certain cellular processes and defining their time evolution of actin engagement remains challenging due to a lack of compatible probes and molecular tools. Here, we developed CASTs, which enable control over ABD activity in response to peptide, small molecule, or light stimuli. CASTs add to a growing toolbox of engineered proteins that can be expressed and subsequently activated to probe the influence of F-actin[76–80]. Being protein-based, these tools offer notable advantages since they are genetically encoded and can be expressed in cells that may be difficult to treat with exogenous probes. Most of the previous tools, however, exert their activity on all actin filaments, either by perturbing polymerization or depolymerization of actin[76,78] or by severing of F-actin[77,79], in turn impacting numerous ABDs. CASTs differ by controlling the binding of a single ABD to F-actin, keeping the actin filament and the other bound ABDs intact and engaged.

The suite of CASTs provides different timescales of activation and the possibility of multiplexing as each CAST module can be triggered orthogonally. Further, orthogonal activation allows control of actin-binding at distinct subcellular locations at different times and in varied order, all within the same cell. With multiple options for activation, CASTs can accommodate a wide variety of experimental demands. Activation deep within organisms can often be challenging with exogenous reagents or light, yet pepCASTs can be activated by a genetically encoded peptide stimulus, enabling control of actin-binding in vivo. optoCAST, on the other hand, is well suited for sequential activation of individual cells during a development process or in subcellular locations in culture. For control over the single and tens of minutes timescale of activation, optoCAST and smCAST will be the most useful, while long timescale, sustained activation will be facile with pepCAST. We showed that CASTs can be inserted into native

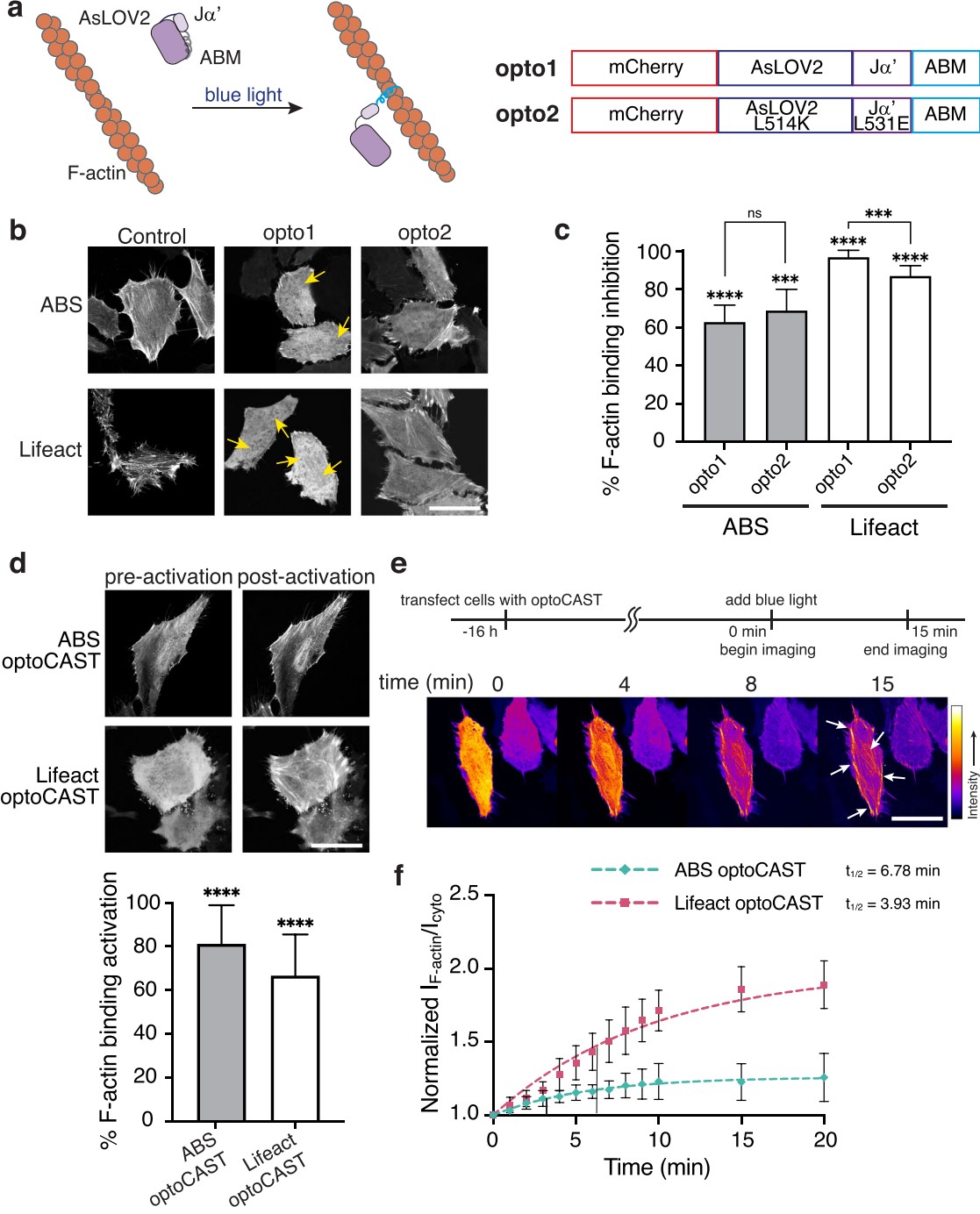

**Fig. 5 | Characterization of optoCAST candidates in cells. a** Schematic of a light-based CAST system. The ABM is initially occluded (gray) by the AsLOV2 domain until blue light illumination leads to the active state (blue) capable of F-actin binding (left). The primary sequence of light-based CAST candidates tested (right). **b** Fluorescent micrographs of fixed HeLa cells expressing opto1-opto2 in the absence of blue light. ABS and Lifeact optoCASTs (opto1) were selected for further study since their localization to F-actin was significantly reduced (yellow arrows). Scale bar 30 μm. **c** Quantification of F-actin binding inhibition for ABS and Lifeact opto1-opto2, where 0% is defined by ABM-only localization and 100% is defined by mCherry-only localization. Bars represent mean ± SD. $n = 20$ biological replicates. **d** Fluorescent micrographs of optoCASTs before and after blue light illumination in live HeLa cells (top). Quantification of activation (bottom), where 0% is defined by inactive optoCAST localization and 100% is defined by ABM-only localization. Scale bar 20 μm. Bars represent mean ± SD. $n = 20$ biological replicates. **e** Time-lapse imaging of Lifeact optoCAST under blue light illumination shows increased localization of optoCAST to F-actin (white arrows) over time. Scale bar 20 μm. **f** Kinetics of ABS and Lifeact optoCAST activation. The ratio of optoCAST's F-actin-localized fluorescence intensity to its cytoplasmic fluorescence intensity normalized to mCherry-only localization is plotted at several timepoints, up to 20 min. $t_{1/2}$ = binding half-time. $n = 3$ biological replicates. Data were fit to a one-phase exponential association model (broken lines). $P$ values were determined using a two-tailed unpaired $t$-test comparison with ABM-only control (**c**) or inactive optoCAST (**d**). (ns not significant $P > 0.05$; *$P < 0.05$; **$P < 0.01$; ****$P < 0.0001$). Source data are provided as a Source Data file.

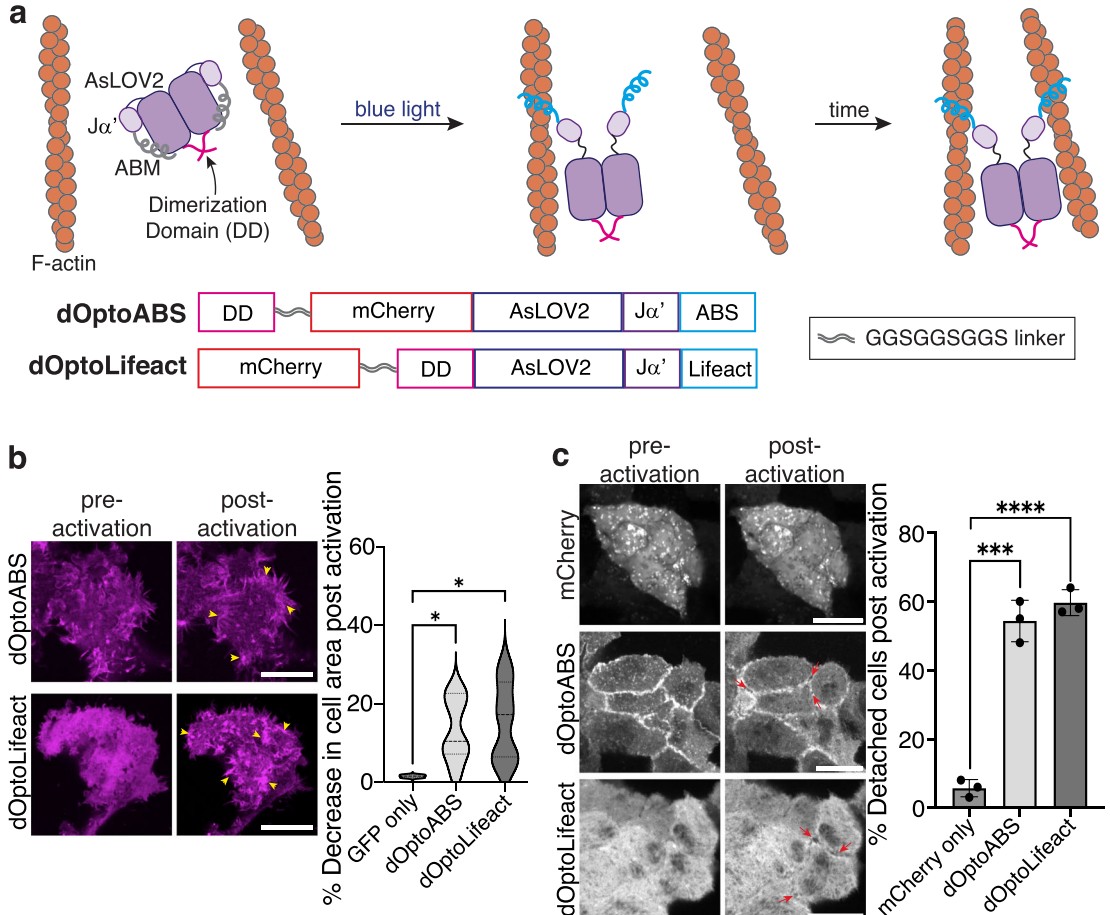

Fig. 6 | Engineering a controllable actin crosslinker with optoCASTs in cells and tissue. a Schematic of the design and activation of a dimeric optoCAST. A self-associating peptide (pink) is tethered to either ABS optoCAST (dOptoABS) or Lifeact optoCAST (dOptoLifeact) to form dimeric versions of optoCAST. Photo-activation of the dimeric species would then expose the two occluded ABMs, leading to actin crosslinking and bundling in cells (top). Primary sequences of dOptoABS and dOptoLifeact (bottom). b Fluorescent micrographs of HEK 293 T cells expressing either dOptoABS or dOptoLifeact before and after blue light illumination (left). Clusters of bundled filaments are present post-activation (yellow arrowhead). Quantification of cell area changes following activation (right). Scale bar 20 μm. n = 12 biological replicates. c Fluorescent micrographs of MDCK II cell islands expressing either dOptoABS or dOptoLifeact before and after blue light illumination (left). Cells contract within the island and detach from one another (red arrows). Quantification shows an increase in cell detachment post-activation (right). Scale bar 15 μm. Bars represent mean ± SD. n = 25 cell-cell contacts. P values were determined using a two-tailed unpaired t-test comparison with GFP-only control (b) or mCherry-only control (c). (ns not significant P > 0.05; *P < 0.05; **P < 0.01; ****P < 0.0001). Source data are provided as a Source Data file.

protein sequences, allowing user-defined manipulation of single protein ABD activity in a background of abundant and diverse ABDs. For this use of CASTs, the target ABD-containing protein must be amenable to protein fusions. We also found that CASTs can be further engineered into synthetic structures, creating architectures that reorganize the cytoskeleton. CASTs present a much-needed strategy for uncovering key aspects of the cytoskeleton and for manipulating cell mechanics and behavior in a user-defined manner.

## Methods
### Cell culture and cell lines
HeLa and HEK 293T cells were a gift from Jeanne Stachowiak (UT Austin), and MDCK II cells were a gift from Keith Mostov (UCSF). All cell lines were maintained at 37 °C and 5% CO$_2$ in high glucose DMEM (4.5 g/l), supplemented with 10% fetal bovine serum (FBS) and 1% penicillin-streptomycin (pen-strep). All transient transfections were performed using FuGENE HD (Promega) at a ratio of 1:1 and 3:1 FuGENE HD:DNA (μL:μg) for HeLa and HEK 293T cells, respectively. For transfections, an eight-well chambered cover glass (Cellvis #C8-1.5H-N) was coated with fibronectin (Sigma-Aldrich #F0895) before cells were seeded and left to grow overnight. The DNA was mixed with

transfection reagent, added to the cells, and then allowed to incubate overnight at 37 °C. Transfection efficiency was confirmed the next day through confocal imaging.

Stable cell lines were generated through lentivirus infection. Briefly, HEK 293T cells were transfected using TransIT-293 (Mirus) according to the manufacturer's instructions with three plasmids, pMD.2 g, p8.91, and the construct-of-interest cloned into a pHR vector. Cells were grown for 3 days, after which media was collected, and the virus was concentrated with Lenti-X (Clontech) according to the manufacturer's instructions. Freshly passaged MDCK II cells or HeLa cells cultured for 24 h were transduced with the collected virus and were grown for 2 days before passaging and removing media. Stable cell lines created with fluorescently tagged proteins were sorted and normalized for expression using a Sony MA900 Cell Sorter at the UT Austin Center for Biomedical Research Support Microscopy and Flow Cytometry Facility. After sorting, cell lines were cultured for 2–3 days and fluorescence was confirmed with confocal imaging.

### General methods
All of the chemical reagents were of analytical grade, obtained from commercial suppliers, and used without further purification unless

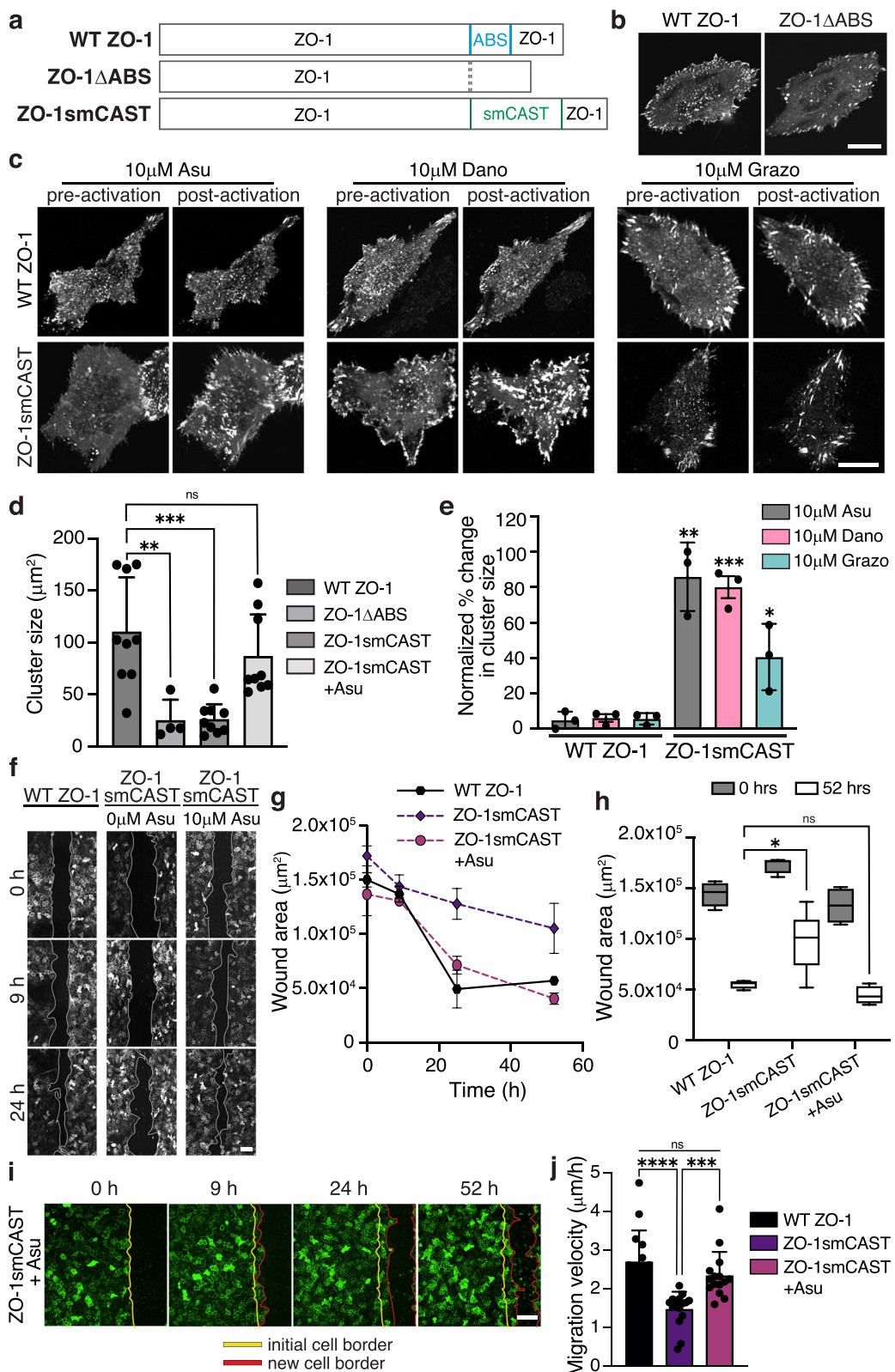

otherwise noted. Alexa Fluor 488 and 647 phalloidin were purchased from Thermo Fisher Scientific. Anhydrotetracycline (ATc) was obtained from Thermo Fisher Scientific (# AAJ66688MA). For smCAST experiments, Asunaprevir (Asu # HY-14434) and Grazoprevir (Grazo #HY-15298) were purchased from MedChem Express, and Danoprevir (Dano #RG7227) was purchased from ApexBio. Stock solutions were prepared and stored according to the manufacturer's instructions.

Working small molecule solutions were obtained by diluting reconstituted stocks in serum-free media.

Fluorescence imaging was carried out on an Eclipse Ti2 microscope (Nikon) equipped with 405/488/560/640 nm lasers (Andor), a Dragonfly 500 high-speed spinning disk confocal module (Andor), and a Zyla 4.2 sCMOS camera (Andor). Fluorescence micrographs of cells were acquired with a 60x objective (Nikon, NA 1.49 TIRF) at 2x camera

**Fig. 7 | Controlling ZO-1-F-actin binding with CASTs modulates cell adhesion and collective migration. a** Primary sequence of full-length WT ZO-1, ZO-1 with its ABS deleted (ZO-1ΔABS), and ZO-1 with its ABS replaced with smCAST (ZO-1smCAST). **b** Fluorescent micrographs of WT ZO-1 and ZO-1ΔABS expressed in live HeLa cells. ZO-1 localizes to focal complexes and large clusters of focal adhesions under the cell body, while ZO-1ΔABS localizes primarily to peripheral focal complexes. Scale bar 20 μm. **c** Fluorescent micrographs of WT ZO-1 and ZO−smCAST before and after Asu, Dano, or Grazo addition in live HeLa cells. After activation, only ZO−smCAST re-localizes to large focal adhesion clusters (lower panel). Scale bar 20 μm. **d** Quantification of ZO-1 cluster size in HeLa cells expressing ZO-1 variants. Bars represent the mean ± SD of clusters. $n = 3$ biological replicates. **e** Change in cluster size of ZO-1 variants before and after drug treatment. Bars represent mean ± SD. $n = 3$ biological replicates. **f** Fluorescent micrographs of collective cell migration in a wounded MDCK II monolayer over time. Stable cells lacking ZO proteins and expressing either WT ZO-1 or ZO-1smCAST were imaged with and without Asu. Scale bar 100 μm. **g** Quantification of wound area over time for cells expressing WT ZO-1 or ZO-1smCAST and cultured in the absence or presence of Asu. Data were presented as mean ± SD. $n = 3$ biological replicates. **h** Quantification of wound area after 52 h. The box represents 25th to 75th percentiles with the middle line as the median and whiskers as the maximum and minimum values. $n = 4$ biological replicates. **i** Time-lapse imaging of wound healing for ZO-1smCAST-expressing MDCK II cells treated with Asu. A leading edge of cells (red line) migrates from an initial wound boundary (yellow line) during wound healing. Scale bar 100 μm. **j** Quantification of the migration rate of cells expressing ZO-1 variants. Bars represent mean ± SD. $n = 15$ biological replicates. $P$ values were determined using a two-tailed unpaired $t$-test. Comparison is to WT ZO-1-expressing cells treated with the same drugs (**e**). (ns not significant $P > 0.05$; *$P < 0.05$; **$P < 0.01$; ****$P < 0.0001$). Source data are provided as a Source Data file.

magnification (unless stated otherwise) and were analyzed using Fiji (ImageJ) and MATLAB. For plate reader assays, sample fluorescence was acquired with a BioTek Cytation 5 Imaging reader. Blue light illumination for in vitro experiments was carried out using the optoWELL LED system from Opto Biolabs GmbH.

## Plasmid construction
All PCR and ligation reactions were performed with OneTaq polymerase and Gibson Assembly Master Mix from New England Biolabs (NEB), respectively. Oligonucleotide sequences used for DNA amplification via PCR are provided in the Supplementary Data 1 file. Vector digestions were performed with HindIII-HF (NEB) and EcoRI-HF (NEB) for pcDNA vectors and MluI (NEB) and NotI (NEB) for pHR vectors. All recombinant DNA was isolated and purified through Miniprep kits according to the manufacturer's instructions (Qiagen). Sequences were confirmed by Sanger sequencing.

## Peptide-based CAST designs
DNA encoding SZ3-linker-ABM-linker SZ4-eGFP sequences was obtained via Gibson Assembly. Designs pep1-9 included a $(GGGGS)_n$ flexible linker on either end of the ABM, where $1 \leq n \leq 9$. For pepCAST activation, SZ21 was fused by Gibson Assembly to mCherry at its C-terminus with flexible linker units between the two fragments. Plasmids encoding SZ 3, 4, and 21 were obtained from Addgene (Addgene plasmid #80659, #80660, and #80675). SZ sequences were amplified via PCR and the fragments were cloned into a digested pcDNA vector backbone. For plasmid constructs with ≥60 total AA residues of the flexible linker, the ABM with the linker residues and Gibson Assembly overhangs was obtained as a g-block from integrated DNA technologies (IDT).

## Small molecule-based CAST designs
Switch modules were designed with NS3A or CP5 on either end of the ABM and with a C-terminal eGFP. Additional variants were designed to include a GGGGS flexible linker on either end of the ABM. Gibson Assembly was used to generate the DNA encoding sequences. Plasmids encoding NS3A and CP5 were obtained from Addgene (Addgene plasmid #133608 and #133614). NS3a variants with D182V and D182Y mutations were obtained as g-Blocks from Integrated DNA Technologies (IDT). Cloning was performed similarly to the peptide-based CAST designs through PCR amplification and Gibson Assembly of the fragments.

## Optical-based CAST designs
Amino acids 403-540 of the AsLOV2 sequence[66] were amplified by PCR. ABMs were fused to the C-terminus of AsLOV2 immediately after AA 540, while mCherry was fused to the N-terminus of the amplified AsLOV2 sequence followed by a flexible linker unit (SRGGSGGSGGSPR). All fragments were joined together by Gibson Assembly (mCherry-LOV-ABM). Variants with L514K and L531E mutations on the AsLOV2 domain were designed similarly[65]. To construct dimeric versions of the optoCAST (dOpto-CAST), a self-recognizing dimerization domain (DD)[81] was appended to the optoCAST sequence. The DD sequence was fused to the N-terminus of mCherry followed by two flexible linker units $(GGGGS)_2$ for dOptoABS (DD-mCherry-LOV-ABS) or placed between mCherry and the LOV domain, also with two flexible linker units, $(GGGGS)_2$, between mCherry and DD for dOptoLifeact (mCherry-DD-LOV-ABS).

The AsLOV2 plasmid was acquired from Addgene (Addgene plasmid #80406). The LOV2 mutant and DD sequences were obtained as g-Blocks from IDT. Cloning was performed similarly to the peptide-based CAST designs through PCR amplification and Gibson Assembly of the fragments.

## ZO-1-smCAST
ZO-1smCAST was obtained by introducing smCAST in place of ABS by making use of a plasmid we previously generated containing ZO-1 with a deleted ABS (ZO-1ΔABS)[33]. For cloning of smCAST into ZO-1ΔABS, a pHR plasmid encoding the ZO-1ΔABS sequence was digested using Ajil (BmgBI) from Thermo Fisher Scientific. The "GACGTG" enzyme recognition sequence was generated during the cloning for truncation of ABS. Cloning steps are similar to those described in the plasmid construction and CAST design sections above with PCR for fragment amplification and Gibson Assembly for ligation.

## Protein expression and purification
DNA sequences for all CASTs (pep, sm, opto) and SZ21 were cloned into a pET28a expression vector containing a 6x His tag either upstream (pepCASTs and SZ21, smCAST) or downstream (opto-CASTs) of the CAST fragment using Gibson Assembly. Plasmids were transformed into *E. Coli* BL21 (DE3) cells and expression was induced by the addition of IPTG after cells reached the desired $OD_{600}$. Specific protein expression conditions for each construct are summarized in Supplementary Table 1. For purification, cells were centrifuged, and cell pellets were resuspended in a lysis buffer containing protease inhibitors (Thermo Fisher Scientific, #A32965). Cells were then lysed first using a Dounce homogenizer followed by a sonication cycle of 2 s on, 2 s off for 2 min and repeated three times in total (Branson SFX250 Sonifier, 30% amplitude). Lysates were then clarified by centrifugation at 14,000×$g$ for 20 min at 4 °C and filtered through a 0.45-μm pore syringe filter. Proteins were purified using affinity chromatography as follows. The clarified lysate was allowed to cycle over a HisTrap FF column (Cytiva #17525501) for 2 h at 4 °C using a BioLogic LP system (Biorad). The column was then washed with wash buffer. For elution, the column was transferred to an Akta Pure 25 (Cytiva), and proteins were eluted using a 20 CV gradient of 0–100% elution buffer. Eluted proteins were further purified by size exclusion chromatography using a Superdex 200 pg column (Cytiva #28989335) into their respective exchange buffers. Eluted proteins were concentrated, aliquoted, and flash-frozen in 10% glycerol for storage at −80 °C. Protein purification was confirmed by SDS-PAGE. Buffer recipes for protein purifications can be found in Supplementary Table 2.

## Fluorescence microscopy

HeLa, HEK 293T, and MDCK II cells expressing plasmids with fluorescent proteins were seeded in eight-well chambered cover glass as previously described in the cell culture section above. The inhibited or activated states were established by acquiring fluorescence images in the absence or presence of SZ21 peptide, small molecule inhibitor, or blue light for the peptide-, small molecule-, and optical-based CASTs, respectively. For quantification of F-actin binding inhibition and activation, CAST-expressing cells in the absence or presence of SZ21 peptide, small molecule inhibitor, or blue light were fixed with 4% PFA (Sigma-Aldrich), permeabilized with Triton X-100 (Thermo Fisher Scientific) and stained with 0.1 μM Alexa Fluor 488 or 647 phalloidin (Thermo Fisher Scientific) for F-actin visualization.

## CAST activation kinetics

**pepCAST**. Stable HeLa cells expressing either ABS pepCAST or Lifeact pepCAST were cultured and seeded in eight-well chambered cover glass as previously described in the cell culture section above. Twenty-four hours after seeding, the cells were co-transfected with a TetR-expressing plasmid and a $TetO_2$-containing SZ21-mCherry plasmid at a ratio of 6:1 (460 ng total DNA) and allowed to incubate overnight. SZ21 expression was induced with the addition of 1 μM ATc, and fluorescence micrographs were acquired every 20 min for 7 h to image pepCAST and SZ21 localization.

**smCAST**. WT HeLa cells were cultured and seeded in eight-well chambered cover glass as previously described in the cell culture section. Twenty-four hours after seeding, the cells were transfected with smCAST and incubated overnight. Next, media was removed from the wells and 10 μM Asu, Dano, or Grazo in serum-free media was added in its place. The well was mounted on the microscope and fluorescence images were acquired every 5 min for 1.5 h to image smCAST localization.

**optoCAST**. WT HeLa cells were cultured and seeded in eight-well chambered cover glass as previously described in the cell culture section. Twenty-four hours after seeding, the cells were transfected with ABS optoCAST or Lifeact optoCAST and incubated overnight. Activation and imaging was performed using the following excitation settings: simultaneous 405 nm (25 mW), 488 nm (55 mW), and 561 nm (55 mW) lasers at 55, 90, and 8% laser powers, respectively. This provided blue light illumination to the cells through the 405 and 488 nm laser lines and allowed visualization of the mCherry-tagged optoCASTs through the 561 nm laser line. Sample irradiation occurred via combined laser line pulses every 2.5 s for the duration of the experiment (8–15 min). For local activation experiments, the illumination aperture was changed from 14.04 × 14.04 mm to 3.25 × 3.25 mm to provide a smaller area of illumination before the activation protocol was run.

## F-actin binding quantification in cells

F-actin binding quantification was performed using Fiji and MATLAB. Briefly, a custom macro script written in Fiji was used to demarcate single cells and separate the CAST candidate channel and the total F-actin channel for each cell. A custom MATLAB code was written to then binarize the total F-actin image and perform background subtraction on the CAST candidate channel. After background subtraction, pixel intensities of the CAST candidate were compared to the binarized F-actin mask to quantify localization to F-actin and the cytoplasm. The following ratio represents the extent of F-actin binding: $\frac{<I_{F-actin}>}{<I_{cyto}>}$. The percent inhibition or activation of each CAST candidate was calculated relative to the intensity ratio of the GFP-only (0%) and GFP-ABM (100%) controls for inhibition or inhibited pep, sm, optoCAST (0%), and GFP-ABM (100%) control for activation. For activation kinetics, F-actin binding quantification was performed as described above but on a frame-by-frame basis. CAST activation

kinetic data was fit to a one-phase exponential association model available with GraphPad. The computed half-time representing the time taken to achieve 50% of the max F-binding ratio is reported as $t_{1/2}$.

## Cell contraction and area change analysis

HEK 293T cells were cultured and seeded in eight-well chambered cover glass as previously described in the cell culture section above. dOpto-CASTs were co-transfected with eGFP at a 1:1 ratio (460 ng total DNA). Photoactivation was performed as described for optoCAST for 10 min. Cell area changes were determined based on the eGFP-only channel before and after activation using Fiji. Stable MDCK II lines expressing dOptoCASTs were generated via Lentivirus and cultured in eight-well chambered cover glass. Photoactivation was performed similarly as in the HEK 293Ts. Analysis of cell monolayer detachment was performed by counting single-cell detachment within clusters of cells.

## ZO-1smCAST activation and localization analysis

WT HeLa cells were cultured and seeded in eight-well chambered cover glass as previously described. Twenty-four hours after seeding, the cells were transfected with WT ZO-1 or ZO-1smCAST and incubated overnight. Activation was performed by the addition of a 10 μM small molecule inhibitor in serum-free media. Fluorescence images were acquired every 5 min for 60 min to observe ZO-1 localization. ZO-1 cluster size was determined using the Fiji plug-in.

## Wound healing assay

WT MDCK II cells and a ZO-1 and ZO-2 null MDCK II cell line (dKO) we previously generated were cultured as previously described[33]. dKO cells were used to generate stable cell lines expressing WT ZO-1 and ZO-1smCAST fusions via Lentivirus. For the wound healing assay, stable cell lines were seeded at a density of 400,000 cells/well on fibronectin-coated eight-well chambered cover glass and incubated overnight to form a monolayer. On the day of the experiment, the cells were washed three times with 1X PBS before a sterile 200 μL tip was used to create a wound in the monolayer by scratching the surface of the well. The wells were washed with 1X PBS to remove unadhered cells and then incubated with serum-free media or serum-free media supplemented with 10 μM of small molecule inhibitor. An image of each well was acquired immediately after wounding, and the cells were incubated at 37 °C until subsequent image acquisitions over the course of 3 days. Images were acquired as a montage and stitched using a Fiji plugin to visualize the entire wound. Analysis of cell migration, including wound area over time and wound healing velocity, was performed using Fiji.

## Actin purification

G-actin was purified from rabbit skeletal muscle according to the protocol from Spudich and Watt[82]. About 20 mL G-buffer (2 mM Tris-HCl, 0.5 mM DTT, 0.2 mM CaCl2, 0.2 mM ATP) was added per gram of ground muscle acetone powder purchased from Pel Freeze (#419952) to a beaker. The sample was stirred on ice for 30 min to begin actin extraction before ultracentrifugation at 4 °C for 30 min at 16,600×g with an S50-A rotor (Thermo Fisher Scientific #45540). The supernatant was filtered through a cheesecloth, then a glass wool layer, and the remaining pellets were resuspended in the same total volume of G-buffer as the supernatant, collected and filtered similarly. The collected supernatants were combined and left to stir gently at room temperature while KCl and MgCl were slowly added to final concentrations of 50 mM and 2 mM, respectively. After 15 min at room temperature, the sample was stirred for 15 min at 4 °C while more KCl was slowly added to a final concentration of 0.8 M. The sample was ultracentrifuged at 4 °C for 2 h at 79,500×g with an S50-A rotor (Thermo Fisher Scientific #45540) to pellet the obtained actin. The supernatant was discarded, and the pellet was carefully transferred to a Dounce homogenizer. About 3 mL G-buffer per gram of muscle

acetone powder prepared was added to the pellet to homogenize it in the Dounce. Actin was then transferred to dialysis tubing (Spectrum™ #132680) and left to dialyze in 800 mL G-Buffer for 2 days with the G-Buffer replaced daily. The dialyzed sample was ultracentrifuged at 4 °C for 2 h at 79,500×$g$ with an S50-A rotor (Thermo Fisher Scientific # 45540), and the top two-thirds of the supernatant was carefully collected. The supernatant was further purified by size exclusion chromatography using a Superdex 200 pg column (Cytiva #28989335) into freshly prepared G-Buffer using the Akta Pure 25 (Cytiva). Fractions excluding the front of the elution peak were pooled, and the final actin concentration was measured at $A_{290}$ using an extinction coefficient of $26,600\,M^{-1}\,cm^{-1}$. Protein purification was confirmed by SDS-PAGE.

### Co-sedimentation actin-binding assay
F-actin was prepared by polymerizing purified G-actin to 90 µM for 1.75 h at room temperature. Various concentrations of F-actin were combined with a constant concentration of each CAST (0.5 µM) in the absence or presence of stimuli (20 µM SZ21, 10 µM Dano, blue light) in their respective exchange buffers. Sub-stoichiometric concentrations of CAST constructs were used in all experiments, such that the assumption of $[F\text{-actin}]_{total} \approx [F\text{-actin}]_{free}$ was valid. Samples were left to incubate in ultracentrifuge tubes (Thermo Fisher Scientific #45235) for 30 min at room temperature. Samples were then spun down at 150,000×$g$ for 30 min at 4 °C to pellet F-actin. The supernatants were then collected, and unbound CAST fluorescence intensities were analyzed using a plate reader. The fraction bound at each concentration of F-actin was calculated using: $1\text{-}(I_x/I_y)$ where $I_x$ is the fluorescence intensity of the supernatant at any concentration of F-actin and $I_y$ is the fluorescence intensity of the supernatant at [F-actin] = 0 µM.

### Pyrene-labeled actin polymerization assay
Pyrene-labeled actin was obtained from Cytoskeleton Inc. (#AP05A). Lyophilized labeled actin was resuspended to 465 µM with cold distilled water and stored per the manufacturer's instructions. For polymerization, a stock solution of 465 µM pyrene-labeled G-actin was diluted with fresh G-Buffer (2 mM Tris-HCl, 0.5 mM DTT, 0.2 mM $CaCl_2$, 0.2 mM ATP) to 100 µM which was further diluted to a working concentration of 10 µM with G-Buffer. The G-actin solution was left on ice to depolymerize for 1 h before ultracentrifugation at 7300×$g$ for 30 min at 4 °C. The supernatant was then collected and pipetted into wells of a black flat-bottom 384-well assay plate (Corning #3821). 0.5 µM activated CASTs were added to the wells to a total volume of 25 µL. For optoCAST activation, 0.5 µM optoCAST was added to a clean microcentrifuge tube and exposed to blue light (Opto Biolabs GmbH) before being added to actin in the wells. Fluorescence measurements were taken every 30 s for 3 min at excitation and emission wavelengths of $360 \pm 20$ nm and $405 \pm 10$ nm, respectively, with a plate reader to establish a fluorescence baseline. After this, 2.5 µL of 10x F-buffer (90% G-Buffer, 500 mM KCl, 10 mM $MgCl_2$, 10 mM EGTA, 100 mM HEPES, 1 mM ATP) was added to each well and mixed by pipetting. Fluorescence measurements were then taken every 30 s for 1 h and $t_{1/2}$ was calculated per ref. 83.

### Protein structure modeling and analysis
Structural predictions for SZ3, SZ4, ZO-1's ABS, and smCAST were obtained via AlphaFold2 modeling[53,54] and visualized on ChimeraX. The top-ranking models (rank 1) were chosen for analysis. To estimate the end-to-end distance of SZ3 and SZ4, a model predicting the SZ3:SZ4 bound state was generated by introducing a very long flexible linker (GGGGS) fused to SZ3 and SZ4 through their C- and N-terminus, respectively. The flexible linker was omitted in the model shown in Fig. 2. Distances were then measured using the distance tool in ChimeraX.

### WLC model
For pepCAST designs, the end-to-end probability distributions for varying residue lengths were determined according to the worm-like chain (WLC) model[43]. The contour length ($l_c$) used in the model was calculated as the product of the number of peptide bonds and the length of a single residue (3.8 Å). The persistence length used was 3.04 Å found for GS linkers[42,43].

### Statistics
Statistical analyses were performed in Prism 9 (GraphPad Software). Specific statistical tests used are noted in figure legends for the corresponding experiments.

### Reporting summary
Further information on research design is available in the Nature Portfolio Reporting Summary linked to this article.

## Data availability
All data supporting the findings of this study are available within the paper and its Supplementary Information. Source data are provided with this paper.

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

## Acknowledgements
This work was supported by grants from the National Institutes of Health (R35GM142941 to B.B.), the National Science Foundation (Grant 2218467 to B.B.), and the Welch Foundation (Grant F-2055-20210327 to B.B.). We thank Daniel Fletcher and Tiama Hamkins-Indik for their helpful discussions.

## Author contributions
U.M.E. and B.B. conceived the idea. U.M.E., H.K., I.R.-V., and Y.W. acquired experimental data, and U.M.E. performed all data analysis. B.B. and U.M.E. designed the experimental strategy, interpreted the data, and wrote the manuscript. B.B. supervised the study. All authors approved the final draft of the manuscript.

## Competing interests
The authors declare no competing interests.
