## [Peer Review File · Nature Communications]

Reviewers' Comments:

Reviewer #2:

Remarks to the Author:

In their revised manuscript, Effiong et al. have substantively addressed the points raised by us, as well as the other reviewers. The control experiments using Asunaprevir-resistant NS3 mutants makes the smCAST data significantly more rigorous. Our minor concern regarding clarification on actin-binding quantification has also been addressed.

Regarding Reviewer #1's comments:

The in vitro experiments adequately clarify concerns about CASTs' actin-binding behavior and impacts on polymerization. Concerns about CAST expression levels relative to F-actin distorting the F-actin/cyto ratio have also been addressed. Finally, all the other concerns regarding quantification and analysis have also been addressed.

Overall, we believe the manuscript is now suitable for publication in Nature Communications.

Reviewer #3:

Remarks to the Author:

The authors have properly and thoroughly addressed my original concerns and suggestions regarding their previous manuscript submitted to Nature Chemical Biology. The study is now of high quality and suitable for publication in Nature Communications.

Reviewer #4:

Remarks to the Author:
